# Decomposition of Rapeseed Green Manure and Its Effect on Soil under Two Residue Return Levels

**Xiaodan Wang [1,2], Hua Ma [1,2], Chunyun Guan [1,2,3,*] and Mei Guan [1,2,3,*]**

[1] College of Agriculture, Hunan Agricultural University, Changsha 410128, China
[2] Hunan Branch of National Oilseed Crops Improvement Center, Changsha 410128, China
[3] Southern Regional Collaborative Innovation Center for Grain and Oil Crops in China, Changsha 410128, China

* Correspondence: guancy2001@hunau.edu.cn (C.G.); gm7142005@hunau.edu.cn (M.G.)

**Abstract:** The overuse of chemical fertilizers has caused various ecological problems in China, necessitating the development of organic alternative fertilizers. There are few studies on the rapidly emerging organic fertilizer rapeseed green manure, despite the importance of understanding its decomposition efficiency and impact on soil. In this study, using plant residue from 14 rapeseed cultivars, we examined the 30-day decomposition changes under conditions A and B (150 and 300 g of plant residue returned, respectively) and detected the effects of their decomposition on soil nitrogen, phosphorus, potassium, and microorganisms. Under condition B, the 30-day cumulative decomposition and nutrient release rates of rapeseed were higher than those under condition A, and the rapeseed decomposition rate exceeded 50% under both conditions, which is similar to results in legume green fertilizers. Moreover, the decomposition of rapeseed green manure significantly increased the soil nutrient content and effectively improved the soil bacterial community structure and diversity relative to the original soil, especially under condition B. *Thiobacillus*, *Azotobacter*, and *Pseudomonas* are bacteria that responded to plant decomposition, and the abundance of the three bacterial genera after plant decomposition was significantly correlated with the plant decomposition traits and soil nutrient content. In conclusion, rapeseed green manure has potential to offset the use of chemical fertilizers, promoting sustainable agricultural development, and this study provides a reference for such green fertilization measures.

**Keywords:** rapeseed; organic fertilizer; nutrients; soil microorganism

## 1. Introduction

Since the reform and opening-up of China in 1978, grain yield has increased from 304.8 million tons to 669.5 million tons in 2020. However, over the same time period, the consumption of chemical fertilizers has also increased nearly six times, from 8.8 million tons to 52.51 million tons [1]. Such excessive use of chemical fertilizer has not only failed to significantly increase crop yields, but has also damaged soil, causing problems such as farmland nutrient loss, soil hardening, and acidification [2–4], which has led to water eutrophication [5], greenhouse gas emissions [6], and other types of environmental pollution, thus seriously affecting the sustainable development of agriculture [7]. In this context, the use of organic fertilizer is essential. In addition to increasing soil fertility [8], organic fertilizer can also improve the metabolic capacity of microorganisms [9], improve soil properties [10], and reduce damage to the ecosystem [11]. Thus, it is an efficient and safe substitute for chemical fertilizer.

Rapeseed is one of the most widely cultivated oil crops [12], and rapeseed oil accounts for half of the edible vegetable oil produced in China [13]. In addition to its use as cooking oil, rapeseed has a spectrum of uses such as animal feed [14,15], vegetable greens [16], a nectar source for bees [17], and even green manure [18]. Rapeseed green manure, as a kind of organic fertilizer, has high dry matter content and is suitable for a wide range of agroecosystems [16,19]. Additionally, the unique glucosinolate hydrolysate in rapeseed

inhibits weeds and soil nematodes [20–22]. Furthermore, rapeseed green manure can be used in winter fallow fields in rotation with rice [23], tobacco [24], and other crops. When the maximum fertilizer effect (i.e., the flowering period) is reached in March every year [25], returning plants to the soil can provide sufficient nutrients for subsequent crops, providing excellent value.

Currently, the research on green manure is mostly focused on legumes, such as alfalfa (*Medicago sativa* L.) [26] and vetch (*Astragalus sinicus* L.) [27], among other species. Such research on rapeseed green manure is less common. Most published work on rapeseed green manure has focused on glucosinolate hydrolysate [20,21] and improving soil fertility [28], failing to clarify the impact on soil microorganisms. Soil microorganisms are important indicators of soil health, as they play an essential role in decomposing organic matter, cycling nutrients, and fertilizing soil [29]. Our previous studies have shown that the decomposition of rapeseed green manure does not change the main composition of soil microorganisms, but only affects their content [30]. However, this past research only analyzed the degradation of a single rapeseed variety under a single condition, which is insufficient for most applications and quite limited in scope. Thus, we selected 14 different green manure rapeseed varieties as a whole to study their effects on soil, which reflects the impact of rapeseed green manure more comprehensively and enriches the current research on rapeseed green manure. In addition, this study utilizing two levels of residue return treatment provides a basis for the rational utilization of rapeseed as green manure, scientific management of nutrients in farmland, and development of its wider application.

## 2. Materials and Methods

### 2.1. Overview of the Experimental Site

The experiment was conducted from 20 March to 19 April 2019, in the Yunyuan base of Hunan Agricultural University, Changsha, Hunan, China (113°7′ E, 28°18′ N) (Figure 1). The experimental field implemented a "rice–oilseed rape" rotation. Table 1 summarizes the climate during decomposition and the soil conditions before decomposition.

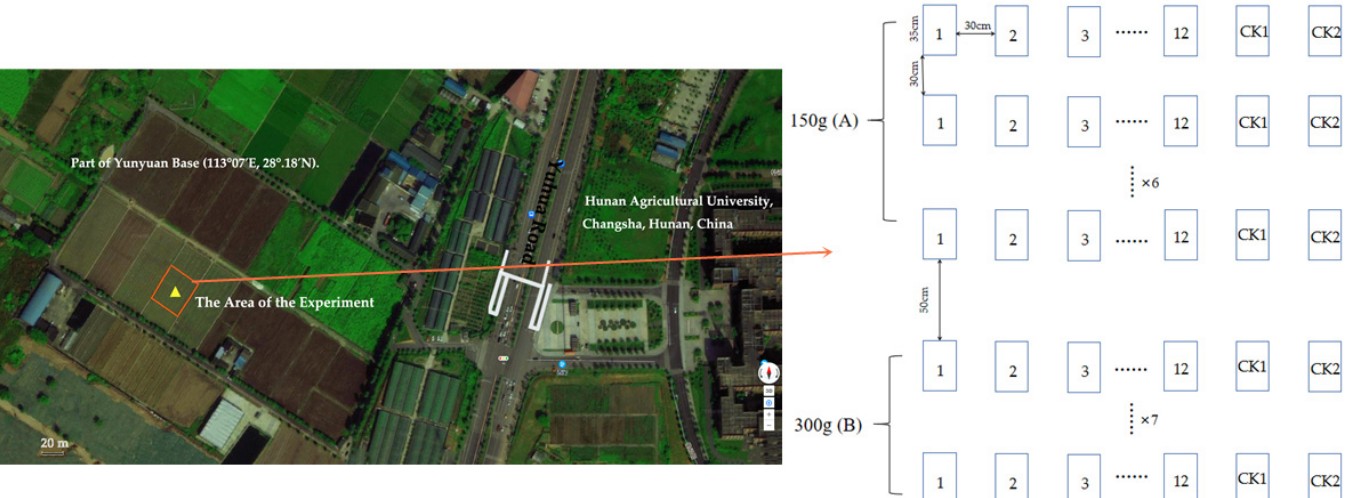

**Figure 1.** Schematic diagram of the experimental site and the experimental design.

**Table 1.** Study site information.

| | Factor Type | Description |
|---|---|---|
| Weather | Climate type | Humid subtropical monsoon climate |
| | Cumulative rainfall | 157.8 mm |
| | Daily average temperature | 16.5 °C |
| Soil | Soil type | Yellow-brown loam soil comprised of 31.5% sand, 38.8% silt, and 29.7% clay |
| | pH | 5.70 |
| | Organic matter | 21.60 g·kg$^{-1}$ |
| | Total nitrogen | 1.29 g·kg$^{-1}$ |
| | Available phosphorus | 10.23 mg·kg$^{-1}$ |
| | Available potassium | 137.76 mg·kg$^{-1}$ |

*2.2. Experimental Design*

This study investigated 14 different rapeseed lines in two residue return treatments, 150 g (A) and 300 g (B), with nine replications (Figure 1). Each residue sample was cut into 2–3-cm segments, mixed well, and packed in mesh bags (25 × 35 cm; 74 μm mesh); they were then arranged in columns keeping the same order in two groups, and buried in 20 cm deep holes spaced 30 cm apart. Plant residues and soils were sampled 10, 20, and 30 days after initiation of the decomposition experiment in three replications. The plant samples were dried to determine the dry plant weight and nutrient content. One part of the soil sample was dried to determine nutrient content, and the other part was flash-frozen in liquid nitrogen and stored at −80 °C for detection of soil microorganisms. Table S1 shows the basic characteristics of rapeseed before decomposition.

*2.3. Test Methods and Data Analysis*

The nitrogen (N), phosphorus (P), and potassium (K) contents of plants and soils were determined using previously defined conventional methods [30–35], and the data were analyzed as previously described [30].

Novogene Biotech Co., Ltd. (Beijing, China) performed the single-end 16S (V3 + V4) regional amplicon sequencing using an IonS5TMXL sequencing platform (Thermo Fisher, Waltham, MA, USA). Bioinformatic analysis of soil microorganisms was conducted with the Novo Magic platform (https://magic.novogene.com (accessed on 13 March 2022), and R (version 4.1.0) and Cytoscape3.7.1 (NIGMS, Bethesda, MD, USA) were used to generate the related figures presented.

Statistical differences between experimental groups were assessed using *t*-tests. SPSS 24.0 (IBM Corp., Armonk, NY, USA) and Origin 2019 (OriginLab, Northampton, MA, USA) were used for correlation analyses and mapping, respectively.

**3. Results**

*3.1. Plant Decomposition and Its Effect on Soil Nutrients under Two Residue Return Levels*

3.1.1. Plant Decomposition under Two Residue Return Levels

The 30-day decomposition rate was essentially identical for conditions A and B. The decomposition rate was the fastest at 10 days, and it decreased at 20 and 30 days. However, the rate under condition B remained higher than that under A (Figure 2a). The cumulative decomposition rates were 51.31% and 53.04% for conditions A and B at 30 days, respectively, but the difference was not statistically significant (Figure 2b).

The total N release rate of rapeseed plants exceeded 30% during the first ten days and gradually decreased until day 30 (Figure 2c,d). At 30 days, the total N released under A and B conditions reached 37.98% and 42.78%, respectively. However, the total P release rate was low, increasing slowly during the first 20 days and decreasing slightly toward 30 days to 18.08% and 20.97% under conditions A and B, respectively. The total K release rate increased rapidly during the first 20 days and leveled off after 30 days. The release rates under A and B conditions were 88.73% and 92.38%, respectively.

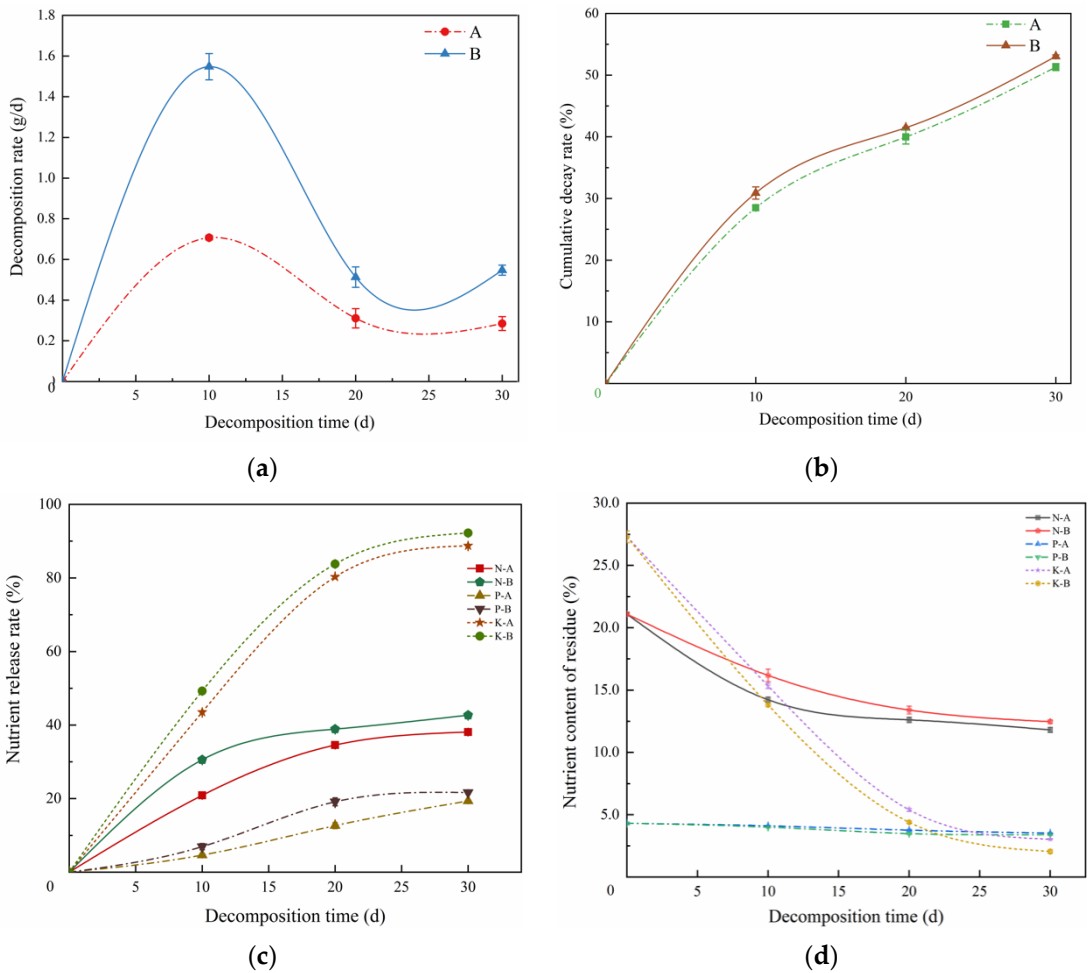

**Figure 2.** Short-term decomposition of rapeseed plants. (**a**) Decomposition speed. (**b**) Cumulative decay rate. (**c**) Nutrient release rate. (**d**) Residual nutrient content.

3.1.2. The Effect on Soil Nutrients under Two Residue Return Levels

After 30 days of decomposition, there was no significant difference in the contents of total N and available P between conditions A and B, while the available K was significantly different ($p < 0.05$) (Figure 3). All the measured parameters were significantly different from the original soil (CK0) under conditions A and B ($p < 0.05$), except the soil total N content under condition A. Moreover, the soil nutrient levels were significantly different among the strains.

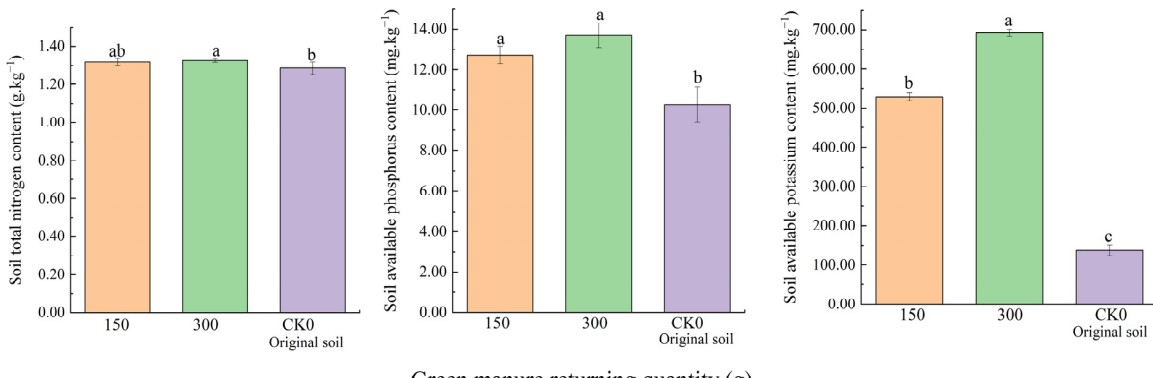

**Figure 3.** Effects of different plant decomposition amounts on soil total N, available P, and available K. Different lowercase letters on graph indicate significant differences ($p < 0.05$).

### 3.2. The Effect on Soil Microorganisms under Two Residue Return Levels

3.2.1. The Effect on Soil Bacterial Community Diversity

After sequence quality control, operational taxonomic unit (OTU) cluster analysis was conducted at a threshold of 97% nucleotide sequence similarity (Figure 4). Thus, 4363 common OTUs were identified from the A, B, and CK0 treatments, accounting for 40.10% of the total OTUs. The control soil (CK0) had 166 specific OTUs, accounting for 1.52% of the total OTUs. These results indicate that rapeseed decomposition substantially changed the soil microbe community. Furthermore, conditions A and B had 1517 and 968 treatment-specific OTUs, respectively, indicating that increasing the amount of materials for decomposition does not necessarily increase soil microbial differences (Figure 4a). Non-metric multidimensional scaling (NMDS) analysis (unweighted) showed that A, B, and CK0 communities were obviously different (Figure 4b). The $\alpha$-diversity of the bacterial community showed a significant difference in community richness among A, B, and CK0 samples ($p < 0.05$), but the community diversity index of CK0 was lower than those of A and B (Table 2).

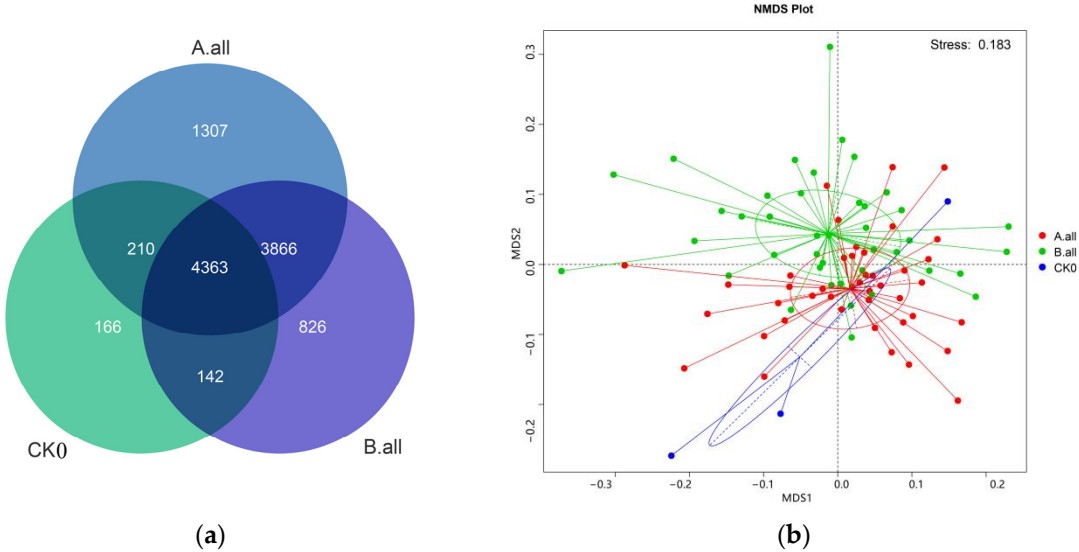

|     |     |
| :---: | :---: |
| (**a**) | (**b**) |

**Figure 4.** Operational taxonomic unit (OTU) analysis among different treatments. (**a**) Venn diagram. (**b**) Non-metric multidimensional scaling (NMDS) plot.

**Table 2.** Bacterial community $\alpha$-diversity under different residue return levels.

|     | Community Richness | | Community Diversity | | Sequencing Depth Index |
| :---: | :---: | :---: | :---: | :---: | :---: |
|     | **Chao1** | **Ace** | **Shannon** | **Simpson** | **Good's Coverage** |
| A | 6528.89 ± 104.37 a | 4289.27 ± 46.46 a | 9.46 ± 0.02 a | 0.9942 ± 0.0009 a | 96.28% |
| B | 4770.01 ± 38.06 c | 3470.84 ± 26.66 c | 9.28 ± 0.02 a | 0.9939 ± 0.0005 a | 97.14% |
| CK0 | 5669.44 ± 217.12 b | 3922.29 ± 105.13 b | 8.99 ± 0.38 a | 0.9815 ± 0.0152 a | 96.68% |

The values with different lowercase letters in the same column are significantly different ($p < 0.05$).

3.2.2. The Effect of Soil Bacterial Content at the Phylum Level

Based on the annotation results using the SILVA database, the top ten bacterial phyla were Proteobacteria, Firmicutes, Acidobacteria, Bacteroidetes, Actinobacteria, Thaumarchaeota, Chloroflexi, Gemmatimonadetes, Verrucomicrobia, and unidentified bacteria, respectively. Nevertheless, CK0 alone and A together with B clustered into two distinct groups, and most strains clustered into two further groups under A and B conditions (Figure 5a), respectively. The principal component analysis (PCA) showed that A, B, and CK0 OTUs were divided into three significantly different groups, indicating that the soils under treatments A and B had unique bacterial phylum characteristics from the original soil (Figure 5b). Specifically, the Proteobacteria abundance under treatment A was signifi-

cantly lower than that under treatment B ($p < 0.05$), but the abundance of Firmicutes under treatment A was significantly higher than that under treatment B ($p < 0.05$). Under A and B conditions compared with CK0 soil, the abundance of Acidobacteria and Bacteroidetes significantly increased ($p < 0.05$), while the abundance of Actinobacteria and Chloroflexi significantly decreased ($p < 0.05$) (Figure 5c). Moreover, the linear discriminant analysis effect size (LEfSe) revealed 42 taxon-specific biomarkers under different residue return levels (Figure 5d).

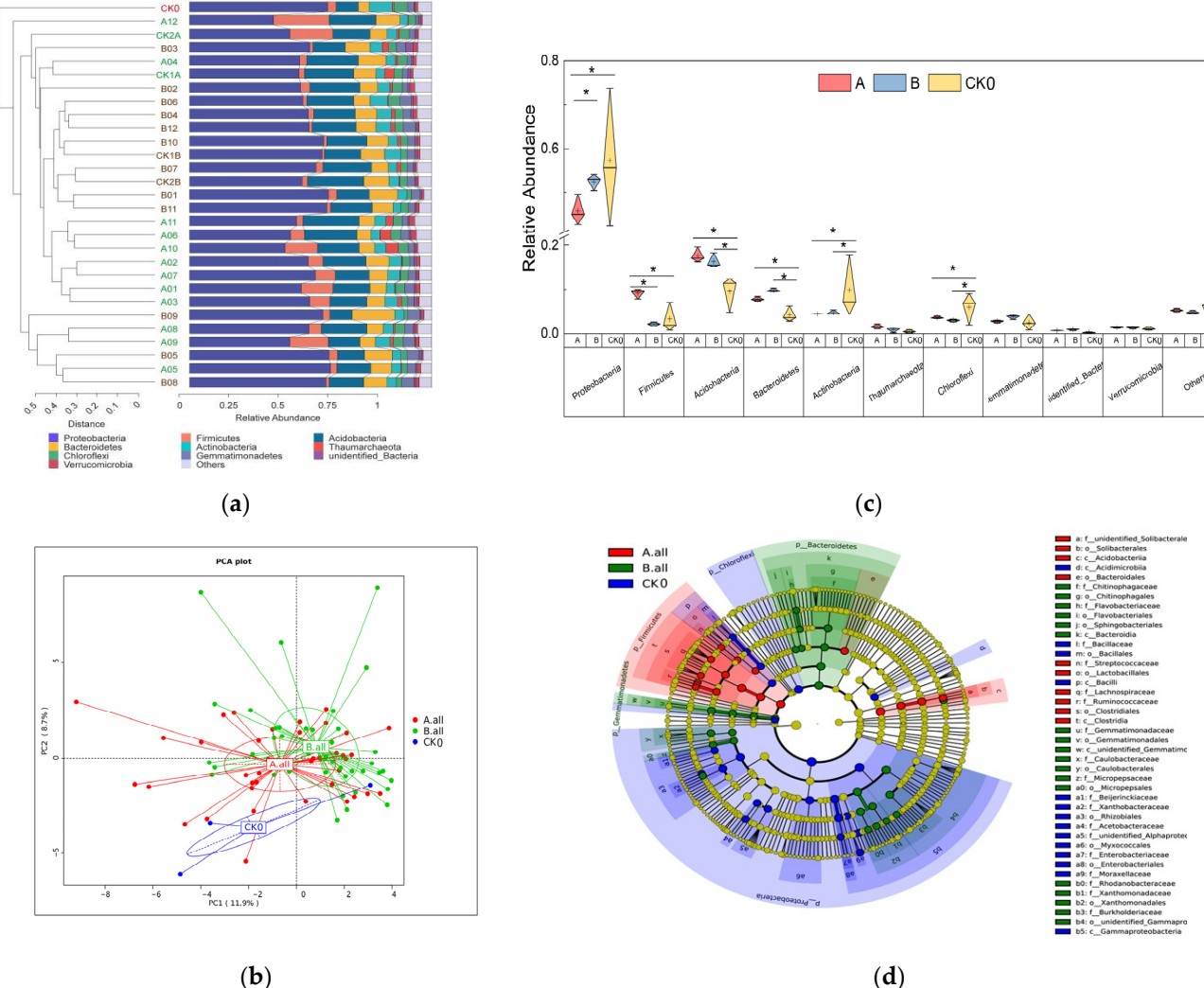

**Figure 5.** Differences in microbial communities at the phylum level among different treatments. (**a**) Unweighted pair group method with arithmetic mean (UPGMA) clustering tree. (**b**) Principal component analysis (PCA) at the phylum level. (**c**) Differences in bacterial community composition among treatments. (**d**) Microbial community cladogram. * indicates a significant difference at the level of $p < 0.05$.

### 3.2.3. The Effect of Soil Bacterial Content and Function at the Genus Level

A clustered heatmap of the top 35 species (Figure 6a) showed that Candidatus Nitrosotalea, Candidatus Solibacter, Acidibacter, unidentified bacteria, Flavisolibacter, Gemmatimonas, Thiobacillus, Geobacter, Tahibacter, Sphingomonas, Bryobacter, Geothrix, and Azotobacter significantly increased ($p < 0.05$) under A and B conditions relative to CK0 soil. In contrast, Enterobacter, Citrobacter, Roseomonas, and Pseudomonas significantly decreased ($p < 0.05$) under conditions A and B. The latter category belongs to Proteobacteria. The functional annotation of prokaryotic taxa (FAPROTAX) analysis showed that the top 35 functions mainly involved nitrogen metabolism (Figure 6b). Moreover, the abundance

of nitrogen metabolism-related genera in the decomposed soil (A and B) was higher than that in CK0 soil. In contrast, there were fewer animal parasitic or symbiotic bacteria and animal and human intestinal microbial flora bacteria. The abundance of soil microbial functional categories differed among decomposed rapeseed lines. However, CK0 and decomposing soils clustered into two branches, indicating that the main functional groups of soil microorganisms changed after decomposition.

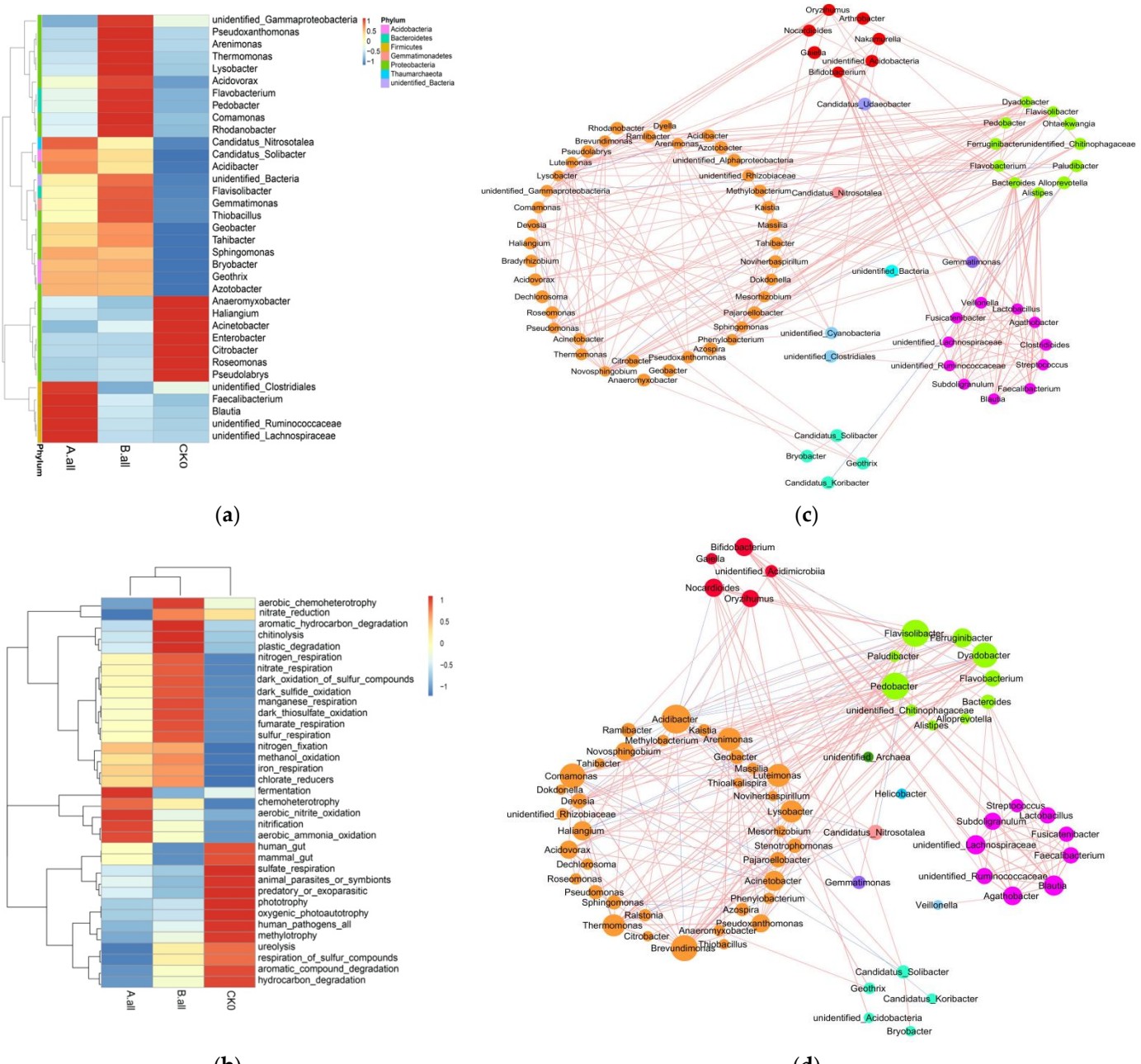

**Figure 6.** Genus-level differences in microbial community among treatments. (**a**) A clustered heatmap of the top 35 genera. (**b**) The FAPROTAX functional results. (**c,d**) The network diagrams of the core genera in treatments A and B, respectively.

We also observed genus-level characteristics from the LEfSe analysis, because the genus is the taxonomic level closest to the ecological functional characteristics. Therefore, the network diagrams of the core genera of A and B were constructed with $|R| \geq 0.65$ and $p < 0.05$ correlation thresholds, respectively (Figure 6c,d). The network relationships of

the two groups were classified into five phyla: *Proteobacteria*, *Actinobacteria*, *Bacteroidetes*, *Firmicutes*, and *Acidobacteria*. The interaction among group A species was strongly positive (more red lines), but the network in group B had higher interconnectivity. Specifically, the two groups shared 126 pairs, while 152 pairs were unique to group A and 103 to group B. Overall, increasing the residue return level did not increase microbial content but enhanced interactions between microorganisms.

### 3.3. Correlation Analysis

The Spearman method was used to analyze the correlations between plant nutrient release rates, soil nutrients, and microorganisms. There were no significant correlations between the plant nutrient release rates and soil nutrients under A and B conditions (Table S2). However, there was a significant positive correlation between the cumulative decomposition rates of plant total N and total P and between dry matter weight, soil total N, and available P (Table S3). Most of the top 35 bacterial genera were significantly correlated with plant dry weight, nutrient release rate, and soil available K (Figure 7), indicating plant decomposition characteristics are an important factor for soil microbial composition.

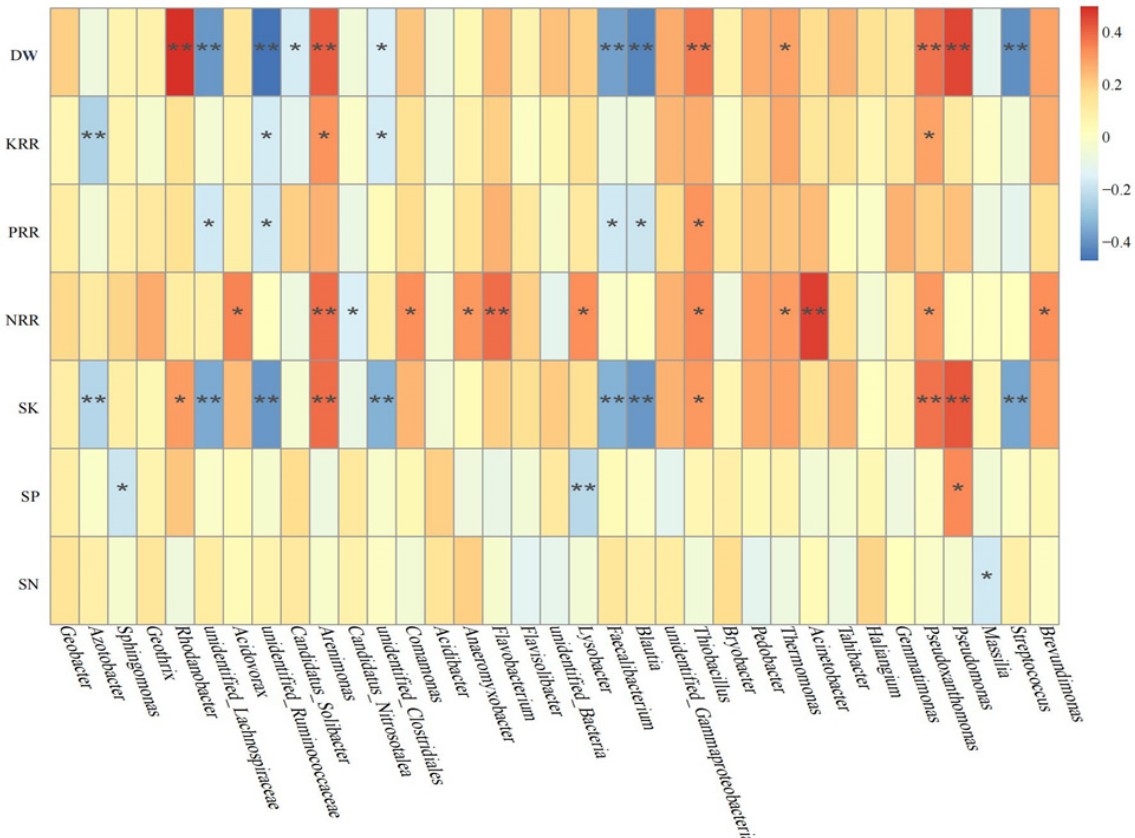

**Figure 7.** Correlations between the top 35 bacterial genera, plant decomposition characteristics, and soil nutrients. DW, dry weight; NRR, PRR, and KRR, the plants' cumulative release rates of nitrogen, phosphorus, and potassium, respectively; SN, SP, and SK, the contents of soil nitrogen, phosphorus, and potassium, respectively. * significant correlation at $p < 0.05$; ** significant correlation at $p < 0.01$.

Moreover, *Thiobacillus* and *Azotobacter* increased significantly after plant decomposition (Figure 6a). Both genera had significant positive and negative correlations with plant decomposition traits, respectively (Figure 7). Meanwhile, *Pseudomonas* significantly decreased after plant decomposition (Figure 6a) but was significantly and positively correlated with plant dry weight decomposition, total N and total K release rates, and soil total N content (Figure 7).

## 4. Discussion

Green manure decomposition is a complex process affected by many factors, including variety characteristics, amount of residue returned, and soil physical and chemical properties [36]. The plant decomposition and nutrient release rate directly reflect the decomposition status after plowing residue under and guide the rational utilization of green manure [37]. In this study, the green manure of 14 different rapeseed varieties was decomposed for 30 days. It was found that the decomposition rate was fast in the first 10 days and slowed down after 20–30 days, both under conditions A and B. However, Liu et al. [25] showed that the rapid decomposition period of rapeseed green manure was 0–28 days. This may be owing to their low temperature in the early stage of the experiment and their use of a pot experiment, in contrast to our field experiment. Nonetheless, the cumulative dry matter decomposing rate in this study was over 50% at 30 days, and Meena et al. [38] showed that the decomposition rate of leguminous green manure, such as soybean and green bean plants, is also 45–60% within the same time span. In this respect, there appears to be little difference between rapeseed green fertilizer and leguminous green fertilizer. In addition, different nutrients have different release rates during decomposition. For example, the release rate of K was significantly higher than that of N and P, consistent with the results of Zhou et al. [39] and He et al. [40]. The release rate of each nutrient is closely related to its existing form. In particular, K exists in its ionic form, which easily dissolves in water; hence, it is released fastest. In contrast, N and P mainly exist in their organic forms, which are not easily decomposed; thus, they are released slowly [39,41]. The cumulative decomposition rate and release of total N, P, and K were higher in condition B than in A. Therefore, increasing the residue return amount can appropriately promote the decomposition of rapeseed green manure, consistent with the results of He et al. [42].

From the perspective of the effect of the amount of plant residue returned on soil nutrients, only soil available K significantly differed between conditions B and A after 30 days of decomposition. Several hypotheses may explain these results. First, this may be explained by the different amounts of plant residue returned. Second, the release rate of total nitrogen and total phosphorus of plants could have been low enough to result in a nonsignificant difference between A and B. Third, the ability of soil to absorb nitrogen and phosphorus may be weaker than its ability to absorb potassium. Further experiments will be required to test these hypotheses. In addition, compared with CK0 soil, all the parameters increased significantly except the soil total N content under condition A, soil K increased by four to six-fold after rapeseed plant decomposition began, in particular, and the contents of soil N, P, and K were highest under condition B. These results show that rapeseed decomposition can increase the soil nutrient content, and increasing rapeseed residue return can indeed increase soil nutrients. Similarly, Zhang et al. [28] showed that when using a rotation crop, rapeseed can reduce the input of N fertilizer for rice. According to the conventional fertilizer standards for rice (N, 150–200 kg·ha$^{-1}$; $P_2O_5$, 90–120 kg·ha$^{-1}$, $K_2O$, 130–160 kg·ha$^{-1}$), this study can reduce the amount of chemical fertilizer by about one third. At the same time, about 50% of the original green fertilizer was still not decomposed by the end of the experiment, and the subsequent decomposition can provide continuous nutrient inputs for rice growth. On the other hand, there was no correlation between soil nutrients and plant decomposition; this may be owing to the short decomposition time, and the bacterial action, pH value, and temperature of soil will also affect plant decomposition and soil nutrient content [43,44], which merits further study.

As an important indicator of soil quality, soil microorganisms play a key role in the soil ecosystem [45,46]. The present study identified Proteobacteria, Firmicutes, Acidobacteria, Bacteroidetes, and Actinobacteria as the dominant soil bacteria after rapeseed decomposition. The LEfSe results further suggested that these bacteria are potentially functional and active soil microbiota in plant decomposition. In addition, OTU cluster analysis and the core genus networks of conditions A and B showed that increasing residue return levels would not cause more microbial changes, but could increase the abundance of beneficial bacteria and enhance community interactions. Under both treatments A and B, the abun-

dance of *Acidobacteria* and *Bacteroidetes* in the soil increased significantly ($p < 0.05$), while the abundance of *Actinobacteria* and *Chloroflexi* decreased significantly ($p < 0.05$) compared to CK0 soil. *Acidobacteria*, as an acidophile, degrades plant residue polymers, enhances photosynthesis, participates in iron cycling and the metabolism of single-carbon compounds, and its relative abundance can also reflect soil acidity [47,48]. The increased relative abundance of *Acidobacteria* in this study suggested that returning rapeseed plant residue to the field possibly lowered soil pH, which can be further detected. Additionally, previous studies have shown that green manure can promote the abundance of *Bacteroidetes* [49,50], which is a sensitive biological index of agricultural soil utilization [51], and this study also found that the higher the oilseed return level within a certain range, the higher the content of *Bacteroides*. On the contrary, *Chloroflexi* [52], similar to *Actinobacteria* [49], registered lower competitiveness than other microbial groups in soils with high organic matter and nutrient content, which explains their decreased abundance after decomposition.

Studies have also shown that one soil type can exhibit different physicochemical properties after different fertilization management methods [53], and different soil physicochemical properties can change the bacterial community structure [54]. Accordingly, we found that most identified microbial groups were significantly correlated with plant decomposition rate, plant nitrogen release rate, and soil potassium content. That is, rapeseed decomposition was mainly responsible for changing the soil environment and recruiting relevant beneficial bacteria. The combined genus-level heatmap and correlation analysis of the top 35 bacterial genera demonstrated that *Thiobacillus*, *Azotobacter*, and *Pseudomonas* changed significantly after decomposition, and they were correlated significantly with plant decomposing traits and soil nutrient content; thus, these genera can be used as indicator taxa for rapeseed decomposition research. We also found that all three genera are *Proteobacteria*, although *Proteobacteria* abundance did not increase significantly. However, as the largest dominant bacterial taxon [55], it is likely important in rapeseed decomposition and merits further research.

**5. Conclusions**

The decomposition of green rapeseed manure can significantly increase the soil nutrient content and effectively improve the bacterial community structure and diversity. Specifically, *Thiobacillus*, *Azotobacter*, and *Pseudomonas* are indicator taxa that responded to plant decomposition and soil nutrient changes and changed significantly after plant decomposition. The combined results from plant decomposition, soil nutrients, and soil bacterial communities showed that treatment B (300 g of residue returned) had the best effect on fertilizing the topsoil. The results of this study may provide some guidance for the appropriate use of green manure. Considering the importance of such potential practical applications, we will further expand the study sample size over various time intervals and in multiple locations as follow-up research to validate these results.

**Supplementary Materials:** The following supporting information can be downloaded at: https://www.mdpi.com/article/10.3390/su141711102/s1, Table S1: Basic characters of rapeseed before decomposition; Table S2: (a) Correlation between plant decomposition characters and soil nutrients under condition A, (b) Correlation between plant decomposition characters and soil nutrients under condition B; Table S3: Correlation analysis of plant and soil nutrient characters before and after decomposition.

**Author Contributions:** C.G. and M.G. designed the experiment. H.M. conducted the experiment, and X.W. processed and analyzed data and wrote the first draft. C.G. and M.G. revised and edited the manuscript. All authors have read and agreed to the published version of the manuscript.

**Funding:** This work was supported by the National Rapeseed Industrial Technology System (CARS-13) and the Natural Science Foundation of China (grant No. 31130040).

**Institutional Review Board Statement:** Not applicable.

**Informed Consent Statement:** Not applicable.



**Data Availability Statement:** All the data included in this study are available upon request by contact with the corresponding author.

**Acknowledgments:** We gratefully acknowledge the help of Bihui Zhang during data analysis, Xiao Ma for his assistance in sample processing, and Tao Chang for helpful scientific discussions.

**Conflicts of Interest:** The authors declare no conflict of interest.

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
