# Peer review of "Decomposition of Rapeseed Green Manure and Its Effect on Soil under Two Residue Return Levels"

_sustainability, doi:10.3390/su141711102_

Round 1
Reviewer 1 Report
Sustainability (ISSN 2071-1050)-MDPI
Manuscript ID: sustainability-1848067
Manuscript Title: "Decomposition of rapeseed green manure and its effect on soil under two returning quantities"
Reviewer comments:
This work described the decomposition of rapeseed green manure and its effect on soil under two returning quantities. The topic is nice and holds practical relevance to green manure and its soil under two returning quantities. On the other hand, the topic fits well with the scope of sustainability-MDPI, and the results are interesting to the scientific community. However, the text needs a major revision before publication. While the innovation is insufficient and some of the discussion is inadequate. It will be deserved a major revision before consideration for publication in Sustainability.
Some of my specific comments are as the below:
· I can see a related full name of B and A was given in the eight and nine sentences of the abstract, but not exactly for B and A. Better to mention the full name of B and A again.
· General Note: The lines in the manuscript are not numbered, which makes it difficult to track notes and changes that will be made in the future in response to reviewers' comments.
· The abstract part is not well written; please write this part according to the scientific writing in this concern. The authors have to follow this sequence; a general introduction about the importance of this study (one sentence), the novelty of the study (one sentence), methodology (one sentence or more but without repetition), and a summary of the most important results, and finally the conclusion and recommendation by simple words.
· Numerical data has been written representing the results obtained and this is good and important in the abstract part of the manuscript, but the authors did not mention a comparison with what?? Please take this note carefully.
· The abbreviation should be used after the full term. Please be consistent with the usage of all abbreviations. Pls, revise the abbreviations in the entire MS.
· There is an extensive presentation of the manuscript problem, please be direct and specific in width and thickness in the form of short and specific sentences that lead you to the goal directly in the abstract section.
· Eight lines without any relevant references “General Note: The lines in the paper are not numbered, which makes it difficult to track notes and changes that will be made in the future in response to reviewers' comments. Pls, add relevant refs.
· Keywords: “Rapeseed green manure; decomposition” I suggest rephrasing some words because keywords should not repeat words from the title.
· The current state of the introduction part needs some more attention.
· Rapeseed oil accounts for half of the edible vegetable oil produced in China 6. Pls add parentheses around the number 6 to follow the journal's instructions in this regard
· In recent years, the multifunctional strategy of developing and utilizing rapeseed has promoted rural revitalization to develop the rapeseed industry 78. What do you mean by 78??
· Besides the extracted oil, rapeseed stalk is a nutritionally rich and delicious vegetable 7. Pls add parentheses around the number 7 to follow the journal's instructions in this regard
· Moreover, rapeseed flowers are colorful and used for sightseeing 910, What do you mean by 910??
· and rapeseed proteins are used to produce vegetable protein fiber and cloth-making 11. The same previous comment
· Yet, rapeseed meal can be used as animal fodder 1213. Rapeseed green manure has high dry matter content and strong adaptability 714. The same previous comment.
· Pls revise the entire manuscript to pay attention to the understanding numbers in the introduction section.
· “Error! Reference source not found” What do you mean by this sentence??
· The plagiarism should be reduced according to the journal's requirements.
· The introduction part is not enough in my opinion. So, pls add more details about the work problem, aim, and avoid repetition.
· Pls add a short paragraph about the importance and novelty of the study compared to previous studies in this regard.
· The materials and methods should be shortened as much as possible in order to reach the intended meaning directly for all the studied characteristics.
· It is important to add relevant and recent references regarding all methods used in this section.
· The results part is long, this part should be shortened as much as possible in order to reach the intended meaning directly for all the studied characteristics
· Figures 6 is not in good displayed, so is it possible to put other colored shapes of better quality and brightness than those listed in the search?
· The authors should take advantage of the chemical analysis of some components they conducted in interpreting their results, especially since these chemical components support a strong discussion of the results if they are used correctly and in the appropriate place.
· “4.1. Decomposition of rapeseed and its effects on soil nutrients under two returning quantity levels” pls delete the titles from the discussion part.
· In the discussion section, conjunctions should be used to show the relationship between sentences.
· Some parts of the discussion sentences need clarification and interpretation, and recent references need to be used as much as possible.
· This part should be better organized and extended. It is important to try to better deepen and explain.
· Conclusions; there are no conclusion parts. There is importance to the presence of this part. In conclusion, you should write a summary of your work in short sentences so that I, as a reader of this article, can understand what the article ended up being
References
· The number of references is about 44 ref. I think it is satisfied. There are a few references for the last three years, why???? So, pls delete the old ones and avoid repetition. There is a recent ref. (2020-2022) in the same trend of the topic of this MS, pls pay attention to this point and cross-check all the references for mistakes, and follow the journal style of reference input.
General comments:
· The manuscript contains some typo errors; please revise it very carefully. A careful revision of the English Grammar is required. So, language needs to be improved thoroughly
· Summarizing, this is a good study, which certainly merits publication after a major revision.

Reviewer 2 Report
Please refer to the attached report

Round 2
Reviewer 1 Report
Sustainability (ISSN 2071-1050)-MDPI
Manuscript ID: sustainability-1848067 (R2)
Manuscript Title: "Decomposition of rapeseed green manure and its effect on soil under two returning quantities"
Reviewer comments:
Dear Editor-in-Chief,
After revising the attached files containing the response to my previous comments on the manuscript, I found that the authors had improved the manuscript and that they responded not completely to most of the comments and questions to be answered.
I think that the manuscript can be accepted for publication after pay attention to the following comments:
- Numerical data has been written representing the results obtained and this is good and important in the abstract part of the manuscript, but the authors did not mention a comparison with what?? Please take this note carefully.
- The introduction part is not enough in my opinion. So, pls add more details about the work problem, aim, and avoid repetition.
- Pls add a short paragraph about the importance and novelty of the study compared to previous studies in this regard.
- In the discussion section, conjunctions should be used to show the relationship between sentences.

Author Response
Thank you for the Again, thank you for your helpful comments.
1.Numerical data has been written representing the results obtained and this is good and important in the abstract part of the manuscript, but the authors did not mention a comparison with what?? Please take this note carefully.
Response 1 : Thank you for reiterating your suggestion, as I believe we missed your meaning in the previous revision. We have now added a sentence in the results of the abstract part, “……which is similar to results in legume green fertilizers.”(line 22).
2. The introduction part is not enough in my opinion. So, pls add more details about the work problem, aim, and avoid repetition.
Response 2 : Thank you again for this comment. we have deleted some unnecessary sentences, and revised the related content. Specifically, we made the following revisions:
The work problem:
- As a new organic fertilizer, the related research on rapeseed green manure is less common ( line59).
- Most published work on rapeseed plants has focused on glucosinolate hydrolysate and improving soil fertility, lacking understanding of the impact on soil microorganisms ( line59-61).
- our previous studies are inadequate and needs more research ( line63 - 68).
Based on these questions,Our purpose is to enrich the research on green manure rapeseed.
3. Pls add a short paragraph about the importance and novelty of the study compared to previous studies in this regard.
Response 3 : In combination with the previous question, the importance and novelty of the study are mainly reflected in:
- Rapeseed green manure needs more research, compared to leguminous green manure (lines 57-59).
- Many different green manure rapeseed (14) as a whole to study their effects on soil, which can reflect the organic fertilizer type of rapeseed green manure more comprehensively and enrich the research on green manure rapeseed (lines 68-70).
- The different residue return treatments can provide a basis for the rational utilization of green manure rapeseed, scientific management of nutrients in subsequent farmland (lines 71-73).
These aspects have been represented in the revised manuscript, Please check again, thank you very much.
4. In the discussion section, conjunctions should be used to show the relationship between sentences.
Response 4 : The use of conjunctive adverbs now makes the relationships between sentences quite clear. Please review again, thank you very much.

Reviewer 2 Report
The Authors improved the article according the suggesytions provided, therefore it can be considered for publication. Still I encourage the Authors to increase the caracter in graphs to make them easier to read if possible
Author Response
As suggested, we have added some information for Figure 1, “…… they were then arranged in columns keeping the same order in two groups,”(line 88). Please review again.
Thank you again for your comments and affirmation.